# Targeted Overexpression of *Claudin 11* in Osteoblasts Increases Trabecular Bone Mass by Stimulating Osteogenesis at the Expense of Adipogenesis in Mice

**DOI:** 10.3390/biology13020108

**Published:** 2024-02-09

**Authors:** Weirong Xing, Sheila Pourteymoor, Anakha Udayakumar, Yian Chen, Subburaman Mohan

**Affiliations:** 1Musculoskeletal Disease Center, Loma Linda VA Healthcare System, Loma Linda, CA 92357, USA; weirong.xing@va.gov (W.X.); sheila.pourteymoor@va.gov (S.P.); anakha.udayakumar@va.gov (A.U.); yian.chen2@va.gov (Y.C.); 2Department of Medicine, Loma Linda University, Loma Linda, CA 92354, USA; 3Graduate School, Loma Linda University, Loma Linda, CA 92354, USA

**Keywords:** Claudin11, overexpression, bone formation, transgenic, adipogenesis, osteogenesis, osteoblasts, skeleton, adipocyte, differentiation

## Abstract

**Simple Summary:**

To interrogate the role of *claudin11* expression in osteoblasts in regulating homeostasis, we generated transgenic mice that express *claudin11* under the control of the rat 2.3 kb *collagen 1α1* promoter. Micro-CT analyses revealed that the distal femoral trabecular bone volume was significantly augmented in the transgenic mice, which was caused by an elevated trabecular number and a reduction in trabecular separation. The increased trabecular bone mass was caused by enhanced bone formation but not by decreased bone resorption in the *claudin11* transgenic mice. The transgenic mice displayed reduced bone marrow adipose tissue and lower expression levels of the adipogenic markers *adiponectin* and *leptin* but higher mRNA levels of the osteogenic markers *Alp* and *Bsp* in the femur. Our data indicate that claudin11 promotes osteogenesis at the expense of adipogenesis in mice.

**Abstract:**

Mice lacking *Claudin11* (*Cldn11*) manifest reduced trabecular bone mass. However, the impact of *Cldn11* expression in osteoblasts in vivo remains understudied. Herein, we generated osteoblast-specific transgenic (Tg) mice expressing *Cldn11* and characterized their skeletal phenotype. Micro-CT analyses of the distal metaphysis of the femur showed a 50% and a 38% increase in trabecular bone mass in Tg male and female mice, respectively, due to a significant increase in trabecular number and a reduction in trabecular separation. Histomorphometry and serum biomarker studies uncovered that increased trabecular bone mass in *Cldn11* Tg mice was the consequence of enhanced bone formation. Accordingly, an abundance of bone formation (*Alp*, *Bsp*), but not bone resorption (*Ctsk*), markers were augmented in the femurs of *Cldn11* Tg mice. Since the trabecular bone density is known to inversely correlate with the amount of marrow adipose tissue (MAT), we measured the MAT in osmium-tetroxide-labeled bones by micro-CT scanning. We found 86% less MAT in the proximal tibia of the Tg males. Consistently, the expression levels of the adipogenic markers, *adiponectin* and *leptin*, were 50% lower in the femurs of the Tg males. Our data are consistent with the possibility that claudin11 exerts anabolic effects in osteoblastic lineage cells that act via promoting the differentiation of marrow stem cells towards osteoblasts at the expense of adipocytes.

## 1. Introduction

Claudin11 (CLDN11) is one of the integral transmembrane proteins and components of tight junction strands that functions as a physical barrier to prevent solutes and water from passing freely through the paracellular space and thus plays a pivotal role in sustaining cell polarity and signal transduction [1,2,3,4]. It is a major component of the central nervous system myelin, which is necessary for central nervous system function, hearing and spermatogenesis [5,6,7]. CLDN11 together with other family member proteins (12–13, 16, 18, 20–24) belong to a group of non-classic claudins based on their sequence similarity [8]. Global knockout (KO) of *claudin11 (Cldn11)* gene in mice results in neurological and reproductive deficits [5]. In addition to the canonical role of CLDNs in forming tight junctions, there is growing evidence that some CLDNs have non-canonical functions in which they participate in intracellular signaling by the phosphorylation of their tyrosine, serine and threonine residues in PDZ binding motifs of the cytoplasmic domain by Rho kinase [9]. The post-translational modifications of CLDNs allow them to interact with other proteins [9]. While several CLDNs are expressed in bone, little is known about their functions in bone metabolism.

In our previous studies, we demonstrated for the first time that CLDN18 was a novel regulator of bone resorption [4]. The disruption of the *Cldn18* gene in mice reduced trabecular bone mass at multiple skeletal sites due to increased osteoclast formation and bone resorption. While mice with global abrogation of the *Cldn11* gene also exhibited a low bone mass phenotype, the trabecular bone volume deficit of the distal femur in the *Cldn11* KO mice was primarily caused by reduced osteoblast differentiation and impaired bone formation via modulating ADAM10-mediated Notch signaling [10]. Our studies suggest that CLDNs function in a cell-type-specific fashion. However, recent studies found that CLDN11 also partially regulated bone mass by negatively regulating RANKL-mediated osteoclast formation in addition to enhancing osteoblast differentiation in vitro and in vivo [11]. The subcutaneous injection of CLDN11 recombinant protein reduced lipopolysaccharide-induced calvarial bone loss and increased calvarial osteogenesis in mouse models [11]. Therefore, the relative contribution of CLDN11 to osteoblast-mediated bone formation vs. osteoclast-mediated bone resorption is unknown. In addition, the mechanism underlying the CLDN11 regulation of osteogenesis needs to be further explored. In this study, we questioned the role of *Cldn11* expression in cells of osteoblast lineage in bone formation in vivo by generating osteoblast-specific transgenic mice expressing the *Cldn11* transgene under the control of the 2.3 kb rat *collagen 1α1* (*rCol1*) promoter and characterized the skeletal phenotype of these transgenic (Tg) mice.

## 2. Materials and Methods

Generation of transgenic mice. To generate Tg lines expressing *Cldn11* in osteoblasts, a full-length mouse *Cldn11* DNA coding sequence was cloned into a modified pWhere vector (Invitrogen) with a 2.3 kb *rCol1* promoter and a chicken *β-actin*/rabbit *β-globin* chimeric intron derived from the ColCAT2.3 plasmid kindly provided by Dr. David Rowe (Center for Regenerative Medicine and Skeletal Development, University of Connecticut, Farmington, CT, USA) [12]. The expression cassette spanning two H19 insulators was released by *PacI* digestion (Figure 1A). Gel purified DNA was microinjected into fertilized *C57BL/6J* mouse ova at the Transgenic Core Facility at the University of Southern California, Los Angeles, California. Genotyping was carried out by PCR with the DNA extracted from the tail tissue. Primer sequences for genotyping were as follows: Forward primer 5′-ATGGTAATCGTGCGAGAGGG-3′; Reverse primer 5′-CCTGAAGGCAAGTGGCTACC-3′. The animal procedure was approved by the Institutional Animal Care and Use Committee (IACUC) of the Jerry L. Pettis Memorial Veterans Affairs Medical Center.

Micro-CT and histomorphometry analyses. The femurs of 13-week-old mice were scanned at 55 kVp for trabecular bone and 70 kVp for the cortical bone with a micro-CT scanner (SCANO Medical, Bruttisellen, Switzerland). For the femoral trabecular bone, the secondary spongiosa region started at 0.36 mm from the distal growth plate in the direction of the metaphysis and extended for 1.89 mm (180 10.5 μm slices). Cortical bones were scanned at the mid-diaphysis of the femur to generate 200 slices (2.1 mm). We integrated the scans into 3-D voxel images and used a Gaussian filter (sigma = 0.8 and support = 1) to reduce the signal to noise ratio. Thresholds of 220 and 260 were applied to all scans for trabecular and cortical bone quantification, respectively, in females (*n* = 8/group) and males (*n* = 10/group). The bone volume (BV, mm^3^), bone volume fraction (BV/TV), trabecular number (Tb.N, mm^−1^), trabecular thickness (Tb.Th, mm) and trabecular space (Tb.Sp, mm) were evaluated. For static histomorphometry, the bones were fixed in 10% formalin in PBS, dehydrated in a series of alcohol solutions and infiltrated with methyl methacrylate. After sectioning, the osteoid was stained using Goldner’s trichrome method. The bone parameters were quantitated by a blinded observer using the OsteoMeasure system (OsteoMetrics, Atlanta, GA, USA).

Enzyme-linked immunosorbent assay (ELISA). The serum carboxy-terminal cross-linked telopeptide of type 1 collagen (CTX-1) and the Procollagen I N-terminal propeptide (PINP) in the Tg and WT male mice (*n* = 10/group) were measured using RatLaps (CTX-1) and Rat/Mouse PINP EIA kits, respectively, according to the manufacturer’s instructions (Immunodiagnostic Systems, Inc., Gaithersburg, MD, USA). Briefly, the strep avidinated microtiter plate was coated with the biotinylated RatLaps Antigen. The plate was then washed and 20 µL of test sera was added to the wells. The plate was incubated together with the primary rabbit polyclonal antibody against the peptide sequence EKSQDGGR at 4 °C. The wells were then washed, and the peroxidase conjugated anti-rabbit IgG was added. The wells were washed followed by the addition of a chromogenic substrate. The color reaction was stopped upon the addition of a stopping solution, H_2_SO_4_, followed by measuring the absorbance reading with a microtiter plate reader. The color intensity was inversely proportional to the concentration of CTX-1 in the serum.

The Rat/Mouse PINP EIA is a competitive ELISA where 50 µL of diluted sample (1:10) was incubated together with a biotinylated PINP reagent in microtiter wells which were coated with a highly specific rabbit polyclonal anti-PINP antibody at room temperature before aspiration and washing. Horseradish peroxidase-labeled avidin was added, followed by washing and the addition of a chromogenic substrate. The absorbance of the stopped reaction mixtures was read with a microtiter plate reader. The color intensity developed was inversely proportional to the concentration of PINP in the serum.

Measurements of marrow adipose tissue and adipocyte counts. The marrow adipose tissue (MAT) was measured using osmium tetroxide, as described previously [13]. Briefly, the tibias from the Tg and control wild-type (WT) mice (*n* = 10/group) were dissected, soft tissues were removed, and the bones were fixed in 4% PFA at 4 °C for 72 h. The bones were then decalcified in 14% EDTA at 4 °C for 18 days. The decalcified bones were labeled in a mixed solution of equal volumes of 2% osmium tetroxide and 5% potassium dichromate for 60 h. After washing, the stained bones were scanned by micro-CT at 55 kVp, with an integration time of 500 ms and a maximum isometric voxel size of 10.5 μm. The volume of the MAT (mm^3^) was quantified at a threshold of 421. The marrow adipocyte counts were measured on the secondary spongiosa region of femur bone sections stained with Goldner with the software of the OsteoMeasure V3.1.0.2 system.

Primary cell cultures. The femurs and the tibias from 4-month-old WT and Tg male mice were cleaned of the muscle and soft tissue; the long bones were then flushed with PBS to remove bone marrow cells. Cleaned bones were cut into small pieces and put in a digestion buffer containing collagenase type II (2 mg/mL), 10% FBS, 100 units/mL of ampicillin and 100 µg/mL of streptomycin in αMEM. Bone fragments were digested at 37 °C in a shaker for 2 h, strained with a 22 µm cell strainer and washed 3 times with PBS. The first digestion was discarded, and the bone chips were digested in a fresh digestion buffer for another 2 h. The digested osteoblasts were then cultured in 10 cm dishes in αMEM containing 10% FBS, 100 units/mL of ampicillin and 100 µg/mL of streptomycin until 70% confluent, followed by RNA extraction. The culturing methods of osteoclast precursor cells derived from the spleen were reported previously [14]. Briefly, the spleens were dissected and placed in αMEM medium in 15 mL plastic tubes on ice. The spleens were then transferred to Petri dishes with 2 mL of αMEM medium and compressed using the plunger of a 3 mL syringe in a circular motion against the Petri dish until no spleen tissue pieces were visible. The homogenized spleen tissues were then transferred into 15 mL plastic tubes, vortexed for 30 s and strained with 22 µm cell strainers. The filtered splenocytes were cultured in αMEM containing 10% FBS, 10% conditional medium derived from an M-CSF-producing cell line, CMG14-12 kindly provided by Dr. Sunao Takeshita as described [15], 100 units/mL of ampicillin and 100 µg/mL of streptomycin. After 3 days of culture, the medium was removed, and the attached osteoclast precursor cells were washed with PBS 5 times, followed by an additional 3 days of culture until the cells were 90% confluent for RNA extraction.

Quantitative real time RT-PCR. Total RNA was extracted from the femurs of the Tg and WT male mice (*n* = 10/group) with the Trizol reagent and was used for the evaluation of gene expression by reverse transcription and real-time PCR as described [16]. The primer sequences are listed in Table 1. Relative gene expression was determined by the ΔΔCT method [17].

Statistical Analysis. The data are presented as the Mean ± SEM (*n* = 8–10) per genotype for each gender and analyzed using the Student’s *T*-test.

## 3. Results

### 3.1. Transgenic Mice Have No Changes in Body Weight and Length but Display a Reduction in Cortical Bone Volume

To examine the role of *Cldn11* expression in osteoblasts, we generated transgenic mice. The transgene, containing two H19 transcriptional insulator element sequences derived from the *Igf2* locus in opposite directions, was designed to express the *Cldn11* transgene in an integration position- and orientation-independent manner (Figure 1A). The expression of transgene is directed by the 2.3 kb *rCol1* promoter and chicken *β-actin*/rabbit *β-globin* chimeric intron. Three Tg mice founders expressed up to 45-fold higher levels of the *Cldn11* transgene in their tail tissue compared to the control WT mice (Figure 1B,C). The first-generation mice (F1) from Tg Line 10 overexpressed up to 70-fold more *Cldn11* in the femur compared to the littermate control WT mice (Figure 1D). Line 10 mice were further bred with *C57BL/6J* mice for more than 10 generations to generate *Cldn11* Tg and WT mice with a pure genetic background and stable expression of *Cldn11* for phenotypic characterization. The abundance of *Cldn11* expression in osteoclast precursors derived from the spleen and osteoblasts derived from the femurs and tibias were 1.8 and 5.7 times higher in the Tg male mice compared to the control WT male mice (Figure 1E). There were no significant differences in body weight, body length or femur length between *Cldn11* Tg and WT mice of either gender (Figure 2A). By contrast, body weight, body length and femur length were reduced in female compared to male mice in both genotypes, as expected. Micro-CT analyses showed that there were no significant differences in cortical bone volume (BV) in either gender of *Cldn11* Tg mice compared to control WT mice (Figure 2B,C). However, the BV adjusted for tissue volume (TV) was significantly less in the *Cldn11* Tg mice of both genders. The BV/TV was decreased by approximately 10% in both male and female Tg mice. While volumetric bone mineral density (vBMD) measurements were less in the *Cldn11* Tg mice compared to those in the control mice, the changes did not achieve the required level of statistical significance.

### 3.2. Cldn11 Transgenic Mice Exhibit an Increased Trabecular Bone Mass

We next measured the trabecular bone parameters at the secondary spongiosa region of bone using micro-CT in the control and *Cldn11* Tg mice. The trabecular BV fraction as well as the BV adjusted for TV at the distal femur metaphysis were increased significantly in both male and female 13-week-old *Cldn11* Tg mice compared to those in the control mice. The femoral trabecular BV was elevated by 50% and 25%, respectively, in the Tg male and female mice (Figure 3A,B). The BV/TV was also 30% and 27% greater in Tg male and female mice compared to that in the corresponding control mice. As expected, the trabecular BV fraction was significantly less in the female mice compared with the male mice in both genotypes. The increase in trabecular bone volume in the *Cldn11* Tg mice was caused by significant increases in both the trabecular number and thickness and a reduction in trabecular separation in both genders. Both the trabecular number and thickness were 20% and 10% more in the Tg male and female mice. The trabecular separation was 10% less in both genders of Tg mice. Accordingly, the trabecular vBMD was also significantly increased in the femur metaphysis of both genders of *Cld11* Tg mice compared to that in the control mice. Consistent with the trabecular femur data, the trabecular bone volume of the proximal tibia metaphysis was increased by 43% and 33% (both *p* < 0.01) in the male and female *Cldn11* Tg mice compared to that in the control mice.

### 3.3. Bone Formation Was Increased but Marrow Adipose Tissue Was Reduced in the Cldn11 Transgenic Mice

To further study the mechanism of CLDN11 action on bone, we performed histomorphometry analyses. The BV/TV, trabecular number, osteoid area, and osteoid volume/tissue volume were 30–40% more in the Tg male mice compared to those in the control mice (Figure 4A,B). The serum level of the carboxy-terminal cross-linked telopeptide of type 1 collagen (CTX-1), a bone resorption marker, was not changed in *Cldn11* Tg male mice (Figure 4C). However, the serum level of procollagen I N-terminal propeptide (PINP), a bone formation marker, was increased by more than 200% in *Cldn11* Tg compared with that in the gender-matched WT mice (Figure 4D). Since bone marrow stromal cells are common precursor cells shared for osteogenesis and adipogenesis and since MAT is known to inversely correlate with trabecular bone mass [18], we next measured MAT in osmium-tetroxide-labeled bones using micro-CT and examined the expression of osteogenic and adipogenic marker genes in bone. We found that the MAT and adipocyte number were reduced by 86% and 68%, respectively, in the proximal tibia of the Tg male mice compared to the gender-matched control mice (Figure 5A,B). Correspondingly, the expression levels of the adipogenic marker, *adpn*, and a white adipocyte-specific cell surface protein, *Asc1*, were reduced by 51% in the femurs of Tg male mice (Figure 5C). The expression levels of *leptin*, a peptide hormone released from adipose tissue that is involved in the regulation of appetite, were approximately 50% less in the femurs of the male Tg mice as compared to the corresponding WT males. The mRNA levels of *adipsin*, a known adipokine that is a regulator of bone marrow fat, were 45% less in the Tg males. The expression levels of bone formation (*Alp*, *Bsp*), but not bone resorption (*Ctsk*), markers were increased in the bones of *Cldn11* Tg male mice (Figure 5D).

## 4. Discussion

In our previous study, we demonstrated that 12-week-old *Cldn11* KO mice manifested a 40% reduction in trabecular bone mass due to the reduced trabecular number and thickness and increased trabecular separation [10]. However, the impact of *Cldn11* expression in osteoblasts in vivo remained undefined. Osteoblasts are known to express high levels of CLDNs, including 1–12, 14 to −20, −22 and −23 [19]. In this study, we hypothesized that the tight junction protein CLDN11 expressed in osteoblastic lineage cells could promote osteogenesis and bone formation via a non-canonical mechanism of regulating stem cell differentiation. To test the hypothesis, we generated Tg mice that overexpressed *Cldn11* in mature osteoblasts under the control of the 2.3 kb *rCol1* promoter. While the 2.3 mouse *Col1* promoter is active in osteoblasts in all bones and in odontoblasts in the teeth of both embryos and postnatal mice and the promoter is also a useful tool to study the function of the transgene in vivo [20], we chose the 2.3 kb *rCol1* promoter to direct transgene expression instead of a 2.3 kb mouse *Col1* promoter for two reasons. First, the 2.3 kb *rCol1* promoter has been characterized extensively and used for both transgenic overexpression and the conditional KO of genes of interest specifically in mature osteoblasts [21,22]. Second, while the regulatory elements of the 2.3 kb promoters are well conserved among mammalian species, the minor sequence differences between rats and mice facilitated the generation of rat-specific primers to genotype the Tg mice [23,24]. As expected, the expression of *Cldn11* driven by the 2.3 kb *rCol1* promoter was 5.7-fold higher in osteoblastic lineage cells in the Tg mice. As reported, the *rCol1* promoter was also slightly active in osteoclast precursors [25]. We found that transgenic overexpression of *Cldn11* specifically in osteoblastic lineage cells increased the trabecular bone volume at the secondary spongiosa of the femur and the tibia by 38–50% but had little or no effect on the cortical vBMD, compared to that in the littermate WT control mice. Histomorphometry and serum biomarker measurements revealed that the increased osteoblast-mediated bone formation, but not the reduced osteoclast-mediated bone resorption, was the cause of increased trabecular bone mass in the *Cldn11* Tg mice. In agreement with the micro-CT and histology data, the bones in *Cldn11* Tg mice transcribed higher levels of bone formation markers of *Alp* and *Bsp*, while the expression levels of the bone resorption marker, *Ctsk*, remained unchanged compared to those in the littermate- and gender-matched control mice. Our data indicate that CLDN11 is a selective positive regulator of trabecular bone but not cortical bone formation, suggesting a different mechanism of CLDN11 action in trabecular bone compared to cortical bone. Our finding that CLDN11 exerts an anabolic effect is consistent with another report [11]. However, our histomorphometry and serum biomarker studies did not support the role of CLDN11 in modulating osteoclast differentiation and/or function.

Bone marrow adipocytes and osteoblasts originate from bone marrow mesenchymal stem cells (e.g., marrow stromal cells), and there is a reciprocal relationship between adipogenesis and osteogenesis because the two types of cells share the same precursors. An imbalance between osteogenesis and adipogenesis could led to osteoporosis, wherein the mesenchymal stem cells are favored to differentiate into adipocytes at the expense of osteoblastic lineage cells [26]. Therefore, we speculated that the increased trabecular bone we observed in the *Cldn11* Tg mice might have resulted from augmented osteoblast differentiation at the expense of adipogenesis. To test this possibility, we measured MAT and quantified marrow adipocytes in the osmium-tetroxide-labeled femur bones by micro-CT. We found that MATs and adipocytes were significantly reduced in the proximal tibia of Tg male mice compared to WT mice. The expression levels of the adipogenic markers, *adiponectin* and *Asc1*, were reduced significantly in the bones of Tg male mice compared to those in the littermate control. Our data are consistent with the possibility that CLDN11 exerts anabolic effects in osteoblasts that act via promoting the differentiation of mesenchymal stem cells towards osteoblasts at the expense of adipocytes. The CLDN11 regulation of hematopoietic stem cell fate is believed to occur via interactions with Tetraspanin 3 and the modulation of Notch signal pathways in osteoblasts because the overexpression of CLDN11 in osteoblasts resulted in the upregulation of Notch targets *Hey1* and *Hey2*, and the ablation of CLDN11 expression in osteoblasts downregulated Notch target expression [10]. Our studies are supported by others’ findings that Notch signaling inhibits the adipogenic differentiation of single-cell-derived mesenchymal stem cell clones derived from human adipose tissue [27]. It is known that canonical Wnt signaling plays important roles in both osteogenesis and adipogenesis. There is evidence that the overexpression of the canonical Wnt ligand Wnt10b promotes higher bone mass, and Wnt10b suppresses the differentiation of preadipocytes and blocks adipose tissue formation [28,29]. Thus, whether CLDN11 regulates marrow mesenchymal stem cell fate via activating Wnt signaling needs to be further explored.

Bone marrow adipocytes are different from the white and brown adipose tissues that exist outside of the marrow cavity and are believed to play a critical role in regulating bone homeostasis [30]. The marrow adipocytes not only have direct contact with bone marrow stem cells but also secrete local adipokines (e.g., adiponectin and leptin proteins) and cytokines (e.g., RANKL) that can have site-specific negative impacts on bone mass and strength. There are two subpopulations of bone marrow adipocytes in mice [31]. The regulated bone marrow adipocytes dispersed among hematopoietic cells in the proximal tibia, distal femur and lumbar vertebrae respond to environmental and physiological stimuli, while the constitutive bone marrow adipocytes localized in the distal tibia and caudal vertebra are relatively stable and suppress local bone formation [30]. It appears that both regulated and constitutive bone marrow adipocytes modulate site-specific bone mass. In our study, we observed a dramatic decrease in MAT in the proximal tibia that was accompanied with augmented trabecular bone mass in *Cldn11* Tg mice, suggesting that CLDN11 regulates bone formation via negatively modulating the abundance of the regulated bone marrow adipocytes.

In this study, we unexpectedly observed a slight decrease in cortical bone volume in both male and female *Cldn11* Tg mice. A possible explanation could be that local bone formation at the diaphysis is regulated via a different mechanism of bone remolding by local regulatory factors and the microenvironment [32]. It is also possible that CLDN regulates endochondral bone formation at the metaphysis but not the endosteal bone formation at the diaphysis. Further studies are needed to interrogate and interpret the mechanism for cortical bone loss at the mid-diaphysis.

The limitation of this study is that the parameters and markers for bone resorption, bone formation and adipogenesis were only measured in male mice since they are influenced by the stage of the estrous cycle in female mice. Our future studies will include both male and female mice and examine whether there are gender-dependent bone and MAT phenotypes, and, if so, whether there is synergistic action between steroid hormones and CLDN11 (e.g., androgen or estrogen and CLDN11) in the bones of *Cldn11* Tg mice.

## 5. Conclusions

In the current study, we generated osteoblast specific Tg mice expressing *Cldn11* and characterized their skeletal phenotype. We found that trabecular bone mass was significantly increased in Tg mice resulting from increased bone formation, while marrow adipogenesis was reduced. Our future work will also test the prediction that CLDN11 exerts anabolic effects in bone by promoting the lineage commitment of bone marrow mesenchymal stem cells towards osteoblasts instead of adipocytes in the bone marrow microenvironment in both male and female mice.

## Figures and Tables

**Figure 1 biology-13-00108-f001:**
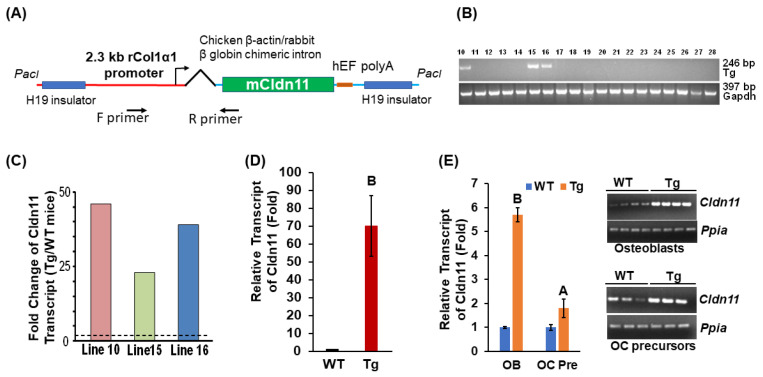
Transgenic mice express high levels of *Cldn11* mRNA in bones. (**A**) A schematic diagram of the expression cassette of the *Cldn11* transgene. hEF, human elongation factor 1. (**B**) Genotyping by PCR identified 3 separate transgenic (Tg) founders. (**C**) The expression levels of the *Cldn11* transcript in the tail tissue of the Tg founders vs. wild-type (WT) 8-week-old mice (WT, *n* = 3). (**D**) The expression levels of *Cldn11* mRNA transcripts in the femurs derived from the first-generation (F1) male mice derived from founder 10 (*n* = 5). (**E**) Stable expression of *Cldn11* in osteoblasts (OBs) derived from the femurs and tibias of the Tg mice with a pure *C57BL/6* genetic background (>10 generations). OC Pre, osteoclast precursors. Images of RT-PCR products on agarose gels are presented in the right panel. The values are the Mean ± SEM. A = *p <* 0.05 vs. control WT mice, B = *p* < 0.01 vs. control WT mice (*n* = 10).

**Figure 2 biology-13-00108-f002:**
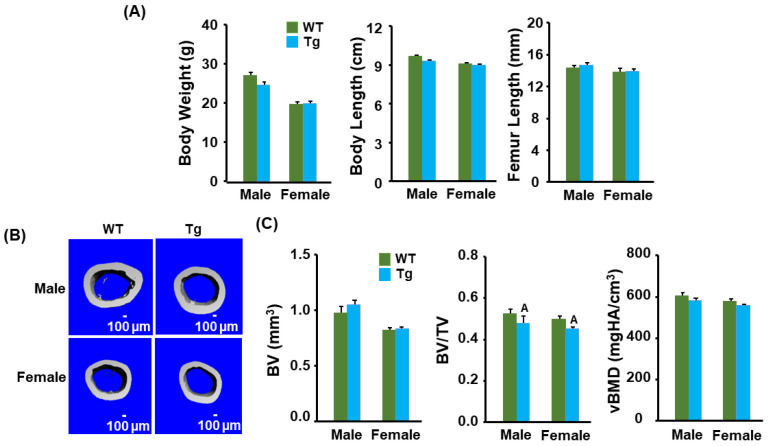
Transgenic mice had no changes in body weight and length but had a reduction in cortical bone volume. (**A**) Body weight, length, and femur length in the *Cldn11* Tg and WT 13-week-old mice. (**B**) Images of femoral cortical bones in 13-week-old *Cldn11* Tg and WT mice. (**C**) Quantitative data of cortical parameters of the femurs of the *Cldn11* Tg and WT littermates. BV, bone volume; TV, tissue volume; vBMD, volumetric bone mineral density. The values are the Mean ± SEM. A = *p* < 0.05 vs. control WT mice (*n* = 8–10).

**Figure 3 biology-13-00108-f003:**
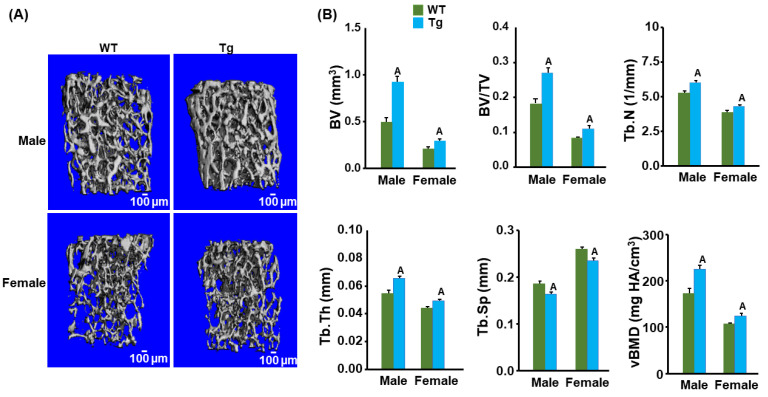
*Cldn11* transgenic mice exhibited increased trabecular bone mass. (**A**) Images of the trabecular bones of the distal femoral metaphysis. (**B**) Trabecular bone parameters of the distal femoral metaphysis in the *Cldn11* Tg and control mice. Tb.N, trabecular number; Tb.Th, trabecular thickness; Tb.Sp, trabecular separation. The values are Mean ± SEM. A = *p* < 0.05 vs. control WT mice (*n* = 8–10).

**Figure 4 biology-13-00108-f004:**
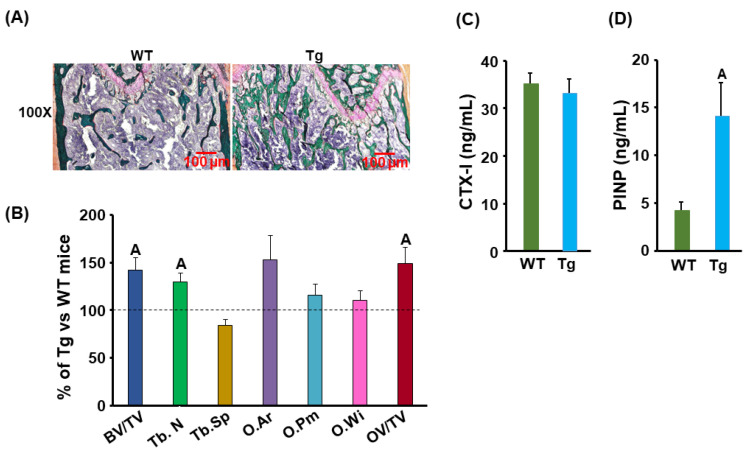
Increased bone formation in the *Cldn11* Tg male mice. (**A**) Images of the femoral sections of the *Cldn11* Tg and WT male mice, stained with Goldner’s trichrome. (**B**) Quantitative data of the femoral sections of the *Cldn11* Tg and WT male mice with the OsteoMeasure software. O. Ar, osteoid area; O. Pm, osteoid perimeter; O.Wi, osteoid width; OV, osteoid volume. (**C**) Serum level of CTX-1, a bone resorption marker, in *Cldn11* Tg and WT mice. CTX-1, carboxy-terminal cross-linked telopeptide of type 1 collagen. (**D**) Serum level of PINP, a bone formation marker, in *Cldn11* Tg and WT male mice. PINP, procollagen I N-terminal propeptide. The values are Mean ± SEM. A = *p* < 0.05 vs. control WT mice (*n* = 10).

**Figure 5 biology-13-00108-f005:**
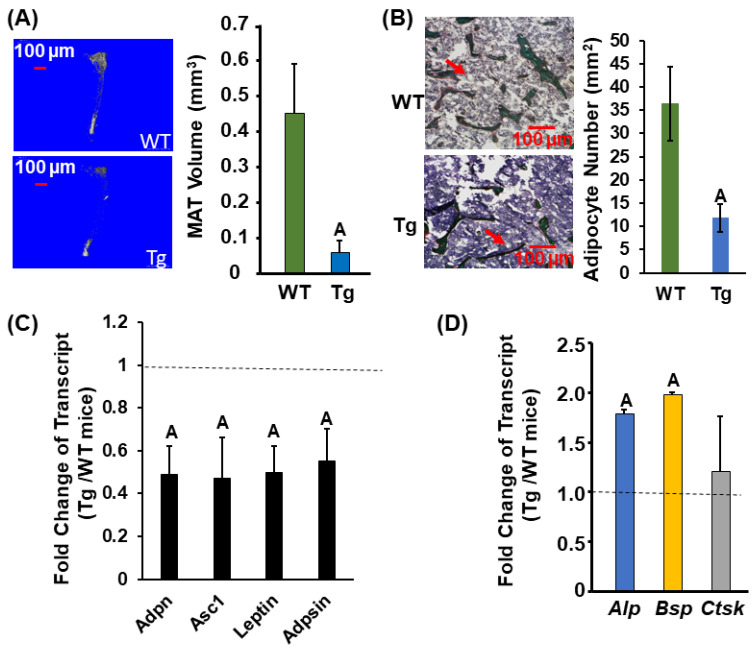
Expression levels of bone formation markers are increased but adipogenic markers are reduced in *Cldn11* Tg male mice. (**A**) Micro-CT images and quantitative data of marrow adipose tissue (MAT) of the tibias in the *Cldn11* Tg and WT male mice. (**B**) Images (100×) and quantitative data of the femurs in the *Cldn11* Tg and WT male mice. The bone sections were stained with Goldner’s trichrome. Red arrows indicate representative adipocytes. (**C**) Expression levels of adipogenic markers in the femurs of the *Cldn11* Tg and WT male mice. (**D**) Expression levels of bone formation (*Alp* and *Bsp*) and bone resorption markers (*Ctsk*) in the femurs, measured by real-time RT-PCR. *Alp*, Alkaline phosphatase; *Bsp*, bone sialoprotein; *Cstk*, cathepsin K.; *Adpn*, adiponectin; *Adpsin*, Adipsin; *Asc1*, alanine/serine/cysteine transporter 1. The values are the Mean ± SEM. A = *p* < 0.05 vs. control WT male mice (*n* = 10).

**Table 1 biology-13-00108-t001:** Primer sequences used for real time RT-PCR.

Gene	Forward Primer	Reverse Primer
*Ppia*	5′-CCATGGCAAATGCTGGACCA-3′	5′-TCCTGGACCCAAAACGCTCC-3′
*Cldn11*	5′- ACCTGCCGAAAAATGGACGA-3′	5′-TGCAGGGGAGAACTGTCAAC-3′
*Alp*	5′-ATGGTAACGGGCCTGGCTACA-3′	5′-AGTTCTGCTCATGGACGCCGT-3′
*Bsp*	5′-AACGGGTTTCAGCAGACAACC-3′	5′-TAAGCTCGGTAAGTGTCGCCA-3′
*Ctsk*	5′-GAACGAGAAAGCCCTGAAGAGA-3′	5′-TATCGAGTGCTTGCTTCCCTTC-3′
*Adpn*	5′-GATGCAGGTCTTCTTGGTCCTA-3′	5′-AGCGAATGGGTACATTGGGA-3′
*Asc1*	5′-GGGTGGCACTCAAGAAAGAG-3′	5′-AGTGTTCCAGGACACCCTTG-3′
*Leptin*	5′-TGCTGCAGATAGCCAATGAC-3′	5′-GAGTAGAGTGAGGCTTCCAGGA-3′
*Adpsin*	5′-GCAGTGGGTGCTCAGTGCT-3′	5′-TCGTCATCCGTCACTCCATC-3′

## Data Availability

The raw datasets generated and/or analyzed during the current study are available from the corresponding author on reasonable request.

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
