# Peer review of "Targeted Overexpression of Claudin 11 in Osteoblasts Increases Trabecular Bone Mass by Stimulating Osteogenesis at the Expense of Adipogenesis in Mice"

_biology, 2024, doi:10.3390/biology13020108_

Round 1

Reviewer 1 Report

Comments and Suggestions for Authors

Overall, the topic and the results in this study are interesting, and the framing of the study looks fine.  However, the preparation of the manuscript is a bit rough and is of lack of rigour.

1.  While it is claimed that the Cldn11 is over-expressed in osteoblasts, the authors have not provided rigorous evidence to confirm this with only expression profiles in the unspecified tail tissue, which might contain other populations of cell types.   Therefore, primary osteoblasts should be isolated to confirm the expression of Cldn11 in this cell type in comparison to the other cell types.

3.  In a number of places, it is just indicated “bone” (such as Figure 1 legend).  The type of bone should be clearly indicated.

4.  In this study, both male and female mice have been used.  But the parameters and markers for bone resorption, bone formation and adipogenesis are only presented for the male mice (Figures 4 and 5).  Why was no data from female mice presented here?

5.  To explain the reduction in cortical bone volume (Lines 245-248), the authors presume the increased secretion of RANKL by MAT.  The authors should provide evidence to support this explanation.  Any study showing the increased RANKL by CLDN11?  The current study has shown the reduction of MAT by CLDN11, which is contrary to the presumed increase in RANKL produced by MAT.

6.  More detail of ELISA method should be given, such as the first and second antibodies and the sources. 

7.  The number of mice of each phenotype and gender should be given in “Materials and Methods”.

8.  Lines 84-85: Directionality of the primers should be given.

9.  The legend for Figure 3 is incomplete.

10. Figure 2 legend: It is better to define the abbreviation in the legend.

11. Minor errors: “B” in Figure 1D and V/T”V” in Figure 3B.

12. English and wording: Please consider correction or change.

     *  Line 45: “together other”

     *  Lines 61-62: “also”…. “also”

 *  Line 114: “real time RT-PCR”

 *  Line 129: “up 45-fold”

     *   Line 146: “as well BV”

Comments on the Quality of English Language

Minor correction needed.

Author Response

Overall, the topic and the results in this study are interesting, and the framing of the study looks fine.  However, the preparation of the manuscript is a bit rough and lacks rigor.

Response: We thank the reviewer for the positive remarks. We also apologize for the rough preparation of the manuscript and lack of rigor.

  1. While it is claimed that the Cldn11 is over-expressed in osteoblasts, the authors have not provided rigorous evidence to confirm this with only expression profiles in the unspecified tail tissue, which might contain other populations of cell types.   Therefore, primary osteoblasts should be isolated to confirm the expression of Cldn11 in this cell type in comparison to the other cell types.

Response: We agree with the reviewer. We have now provided the claudin 11 expression levels in primary osteoblasts as well as in bone marrow stromal cells and osteoclast precursor cells  in the revised manuscript by real time RT-PCR. These data are included in figure 1E. We also specify the bone used in figure 1C, which is femur.

  1. In a number of places, it is just indicated “bone” (such as Figure 1 legend).  The type of bone should be clearly indicated.

Response: We apologize for not specifying the bone. We have now specified the bone as the femur in the revised manuscript.

  1. In this study, both male and female mice have been used.  But the parameters and markers for bone resorption, bone formation and adipogenesis are only presented for the male mice (Figures 4 and 5).  Why was no data from female mice presented here?

Response: We apologize for not analyzing the female Tg mice in this study since it is known that variation in the stage of estrous cycle influence bone turnover.  We have now indicated the limitation of this study in the revised discussion stating “The limitation of the study is that the parameters and markers for bone resorption, bone formation and adipogenesis were only measured in male mice since they are influenced by the stage of estrous cycle in female mice.  Our future study will include both male and female mice and examine whether there is a gender-dependent bone and MAT phenotypes, and if so, whether there is a synergistic action between steroid hormone and CLDN11 (e.g., androgen or estrogen and Cldn11) on bone in the Cldn11 Tg mice.”

  1. To explain the reduction in cortical bone volume (Lines 245-248), the authors presume the increased secretion of RANKL by MAT.  The authors should provide evidence to support this explanation.  Any study showing the increased RANKL by CLDN11?  The current study has shown the reduction of MAT by CLDN11, which is contrary to the presumed increase in RANKL produced by MAT.

Response: We apologize for the inappropriate interpretation. We have now removed the sentences and stated that “The possible explanation could be that bone formation at the diaphysis is regulated via a different mechanism of bone remodeling by local regulatory factors and microenvironment [27]. It is also possible that CLDN regulates endochondral bone formation at the metaphysis but not the endosteal bone formation at the diaphysis. Further studies are needed to interrogate and annotate the mechanism for cortical bone loss at the mid-diaphysis.”

  1. More detail of ELISA method should be given, such as the first and second antibodies and the sources. 

Response: We apologize for not providing the detailed methods of ELISA. We have now described the method in detail in the revised manuscript.

  1. The number of mice of each phenotype and gender should be given in “Materials and Methods”.

Response: We have now provided mouse number and gender in each experiment in the materials and methods in the revised manuscript.

  1. Lines 84-85: Directionality of the primers should be given.

Response: We apologize for missing primer direction. We have not provided the directionality of the primers in the revised manuscript.

  1. The legend for Figure 3 is incomplete.

Response: We apologize for missing a part of the legend due to the figure moving. We have now provided the complete legend in the revised manuscript.

  1. Figure 2 legend: It is better to define the abbreviation in the legend.

Response: We have now defined the abbreviations in the revised legend.

  1. Minor errors: “B” in Figure 1D and V/T”V” in Figure 3B.

Response: We have corrected the error in the revised manuscript.

  1. English and wording: Please consider correction or change.

     *  Line 45: “together other”

     *  Lines 61-62: “also”…. “also”

 *  Line 114: “real time RT-PCR”

 *  Line 129: “up 45-fold”

     *   Line 146: “as well BV”

Response: We have corrected these errors in the revised manuscript.

Reviewer 2 Report

Comments and Suggestions for Authors

To interrogate the role of claudin11 expression in osteoblasts in regulating bone homeostasis, the authors generated transgenic mice that express claudin11 under the control of rat 2.3 kb collagen 1α1 promoter. The manuscript by Xing et al claims that claudin11 promotes osteogenesis at the expense of adipogenesis in mice. However, the manuscript could be improved after major revision if the following comments and questions are addressed.

1.      Why chose rat 2.3 kb collagen 1α1 promoter not mice?

2.      The expression of claudin11 should be detected in different tissues and organs through both qPCR and Western Blot analysis.

3.      Representative calcein double-labeling images showing bone formation capacity should be added.

4.      The results of the primary osteoblast isolated from WT and TG mice should be added.

Comments on the Quality of English Language

The manuscript should be thoroughly edited for grammar. It seems that the manuscript was written in parts, where some sections are with correct English, while other sections were difficult to read. Overall poorly edited manuscript including several typographical errors in scientific writing as well. For example, line 10 “regulating home homeostasis” bone homeostasis.

Author Response

To interrogate the role of claudin11 expression in osteoblasts in regulating bone homeostasis, the authors generated transgenic mice that express claudin11 under the control of rat 2.3 kb collagen 1α1 promoter. The manuscript by Xing et al claims that claudin11 promotes osteogenesis at the expense of adipogenesis in mice. However, the manuscript could be improved after major revision if the following comments and questions are addressed.

  1. Why chose rat 2.3 kb collagen 1α1 promoter not mice?

Response: While the 2.3 mouse Col1 promoter is active in osteoblasts in all bones and in odontoblasts in teeth of both embryos and postnatal mice and the promoter is also a useful tool to study the function of transgene in vivo [18], we chose the 2.3 kb rat col1 promoter to direct transgene expression instead of a 2.3 kb mouse collagen 1α1 promoter for two reasons. First, the 2.3 kb rat Col1 promoter has been characterized extensively and used for both transgenic overexpression and conditional KO of genes of interest specifically in mature osteoblasts in mice [19, 20]. Second, while the regulatory elements of the 2.3 kb promoters are well conserved among mammalian species, the minor sequence differences between rats and mice facilitated generation of rat specific primers to genotype the transgenic mice [21, 22]. We have now added these sentences in the revised discussion.

  1. The expression of claudin11 should be detected in different tissues and organs through both qPCR and Western Blot analysis.

Response:  Unfortunately, we did not save different organs for the analyses, and it will take considerable amount of time to generate more mice for these analyses. However, we analyzed cldn11 gene expression by real time RT-PCR in primary osteoblasts and osteoclast precursors from the mouse spleen. The data are presented in Figure 1E, showing an 5.7-fold increase in expression of cldn11 in osteoblasts derived from mice that have been bred for more than generations to obtain the transgenic line with a pure genetic in C57BL/6 background with stable expression. The expression of Cldn11 is reduced in osteoblasts as compared to bones derived from the F1 Tg mice had multiple integration sites of chromosomes in Figure 1C, D. We have addressed this issue in the revised manuscript.

  1. Representative calcein double-labeling images showing bone formation capacity should be added.

Response: We thank the reviewer for this comment.  Unfortunately, we did not administer fluorescent labels to monitor bone formation in mice.  We, however, performed static histomorphometry and the data were presented in Figure 4

  1. The results of the primary osteoblast isolated from WT and TG mice should be added.

Response:  This data is now added in Figure 1E in the revised manuscript.

Round 2

Reviewer 1 Report

Comments and Suggestions for Authors

A few issues below to be addressed:

1.  Method for preparation of spleen pre-osteoclast should be added.

2.  Table 1: 3’ for the primers should be added.

3.  Line 143: “The disgusted cells …..”.

Comments on the Quality of English Language

No comment.

Author Response

Method for preparation of spleen pre-osteoclast should be added.

Response: We apologize for not providing the detail osteoclast precursor culture.  I have now added the method of osteoclast precursors culture in the revised manuscript (see lines 146-157). 

Table 1: 3’ for the primers should be added.

Response: We thank the reviewer. We have now added 3' in the revised table 1.

Line 143: “The disgusted cells …..”.

Response: We apologize for the typo. We have now corrected the typo and replaced with "The digested osteoblasts...."

Reviewer 2 Report

Comments and Suggestions for Authors

Accept in present form

Comments on the Quality of English Language

Accept in present form

Author Response

We thank the reviewer for positive remarks.